# RNA-seq and ChIP-seq Identification of Unique and Overlapping Targets of GLI Transcription Factors in Melanoma Cell Lines

**DOI:** 10.3390/cancers14184540

**Published:** 2022-09-19

**Authors:** Matea Kurtović, Nikolina Piteša, Nenad Bartoniček, Petar Ozretić, Vesna Musani, Josipa Čonkaš, Tina Petrić, Cecile King, Maja Sabol

**Affiliations:** 1Division of Molecular Medicine, Ruđer Bošković Institute, 10 000 Zagreb, Croatia; 2The Garvan Institute of Medical Research, 384 Victoria St., Darlinghurst, NSW 2010, Australia; 3The Kinghorn Centre for Clinical Genomics, 370 Victoria St., Darlinghurst, NSW 2010, Australia; 4School of Biotechnology and Biomolecular Sciences, Faculty of Science, University of New South Wales, Sydney, NSW 2052, Australia

**Keywords:** melanoma, HH-GLI pathway, GLI targets, MAPK pathway, targeted therapy

## Abstract

**Simple Summary:**

The majority of melanomas show hyperactivation of the MAPK signaling pathway, most often through mutations in *BRAF* and *NRAS.* Despite significant progress in therapy, targeting this signaling pathway solely has not been the solution for stopping the progression of this disease. Recently, researchers recognized the involvement of the Hedgehog-GLI (HH-GLI) signaling pathway in melanoma and its crosstalk with the MAPK pathway. In order to identify new HH-GLI-regulated targets that could be involved in the crosstalk with the MAPK pathway, we performed RNA sequencing and ChIP sequencing on three melanoma cell lines. By combining RNA-seq and ChIP-seq results, we successfully validated 15 novel targets of GLI proteins in melanoma cell lines. These findings will contribute to a better understanding of the GLI code and its role in melanoma.

**Abstract:**

Background: Despite significant progress in therapy, melanoma still has a rising incidence worldwide, and novel treatment strategies are needed. Recently, researchers have recognized the involvement of the Hedgehog-GLI (HH-GLI) signaling pathway in melanoma and its consistent crosstalk with the MAPK pathway. In order to further investigate the link between the two pathways and to find new target genes that could be considered for combination therapy, we set out to find transcriptional targets of all three GLI proteins in melanoma. Methods: We performed RNA sequencing on three melanoma cell lines (CHL-1, A375, and MEL224) with overexpressed GLI1, GLI2, and GLI3 and combined them with the results of ChIP-sequencing on endogenous GLI1, GLI2, and GLI3 proteins. After combining these results, 21 targets were selected for validation by qPCR. Results: RNA-seq revealed a total of 808 differentially expressed genes (DEGs) for GLI1, 941 DEGs for GLI2, and 58 DEGs for GLI3. ChIP-seq identified 527 genes that contained GLI1 binding sites in their promoters, 1103 for GLI2 and 553 for GLI3. A total of 15 of these targets were validated in the tested cell lines, 6 of which were detected by both RNA-seq and ChIP-seq. Conclusions: Our study provides insight into the unique and overlapping transcriptional output of the GLI proteins in melanoma. We suggest that our findings could provide new potential targets to consider while designing melanoma-targeted therapy.

## 1. Introduction

Melanoma is known as the most aggressive and deadliest of all skin cancers. The most often dysregulated signaling pathway in melanoma is RAS/RAF/MEK1/2-ERK1/2 (MAPK pathway). Almost 50% of all melanomas have a mutation in the *BRAF* gene, while 15–20% have a mutation in the NRAS gene [1], which leads to constitutive pathway activation [2]. Still, targeting this signaling pathway solely has not been the solution for stopping the progression of this disease. Due to the low response rates of melanoma patients to targeted therapy and immunotherapy, novel treatment strategies are needed. Recently, researchers shifted their focus to combination therapies and targeting other signaling pathways that are in crosstalk with the MAPK pathway. One of the pathways that are reported to have consistent crosstalk with the MAPK pathway is Hedgehog-GLI (HH-GLI) signaling pathway [3], making it a potential new strategy for melanoma therapy improvement. Abnormal HH-GLI pathway activation has been described in a variety of human cancer types, including medulloblastoma, pancreatic, prostate, colon, breast, ovarian, and lung cancer [4,5,6,7,8,9,10]. The importance of the HH-GLI signaling pathway in melanoma and its resistance to therapy has also been noticed and reported. For example, studies show that inhibition of the HH-GLI pathway can suppress the growth of melanoma cells in vitro and in vivo. Furthermore, it has been demonstrated that GLI downregulation induced apoptosis and that this may contribute to the increased sensitivity of melanoma cells to vemurafenib [11,12,13,14]. However, identifying GLI transcriptional targets in melanoma can provide insight into the role of HH-GLI signaling in the pathogenesis of this tumor. The main effectors of HH-GLI signaling are GLI transcription factors (GLI1, GLI2, and GLI3). They can act as transcriptional activators or repressors; GLI2 and GLI3 harbor an N-terminal repressor domain and can act as both activators and repressors of the pathway, while GLI1, lacking this domain, acts only as a transcriptional activator [15]. In addition, one study identified GLI3 as an effector of KRAS/PI3K/AKT1 signaling in cancer cells [16]. There are two types of HH-GLI pathway activation: canonical and non-canonical. MAPK and PI3K can non-canonically activate the HH-GLI signaling pathway at the level of GLI transcription factors [3,17]. Previous research has already shown that MEK1/2-ERK1/2 signaling acts upstream of HH and regulates the activity of GLI transcription factors. For example, NRASQ61K and HRASV12G improve GLI1 function, increasing its transcriptional activity and nuclear localization [18]. Surprisingly, there are also studies reporting that the upstream members of the MAPK cascade, the mitogen-activated kinases MEKK1 and MEKK2/3, can negatively regulate GLI1 in medulloblastoma cells [19]. In order to find new GLI transcriptional targets that could be considered for combination therapy of melanoma, we performed RNA sequencing on melanoma cell lines with overexpressed GLI1, GLI2, and GLI3 and coupled the data with ChIP sequencing results on endogenous GLI1, GLI2, and GLI3 proteins for additional confirmation of direct GLI targets. Using these two methods, we have been able to confirm and validate 15 novel GLI targets that are involved in MAPK and many other signaling pathways, as revealed by pathway enrichment analysis.

## 2. Materials and Methods

### 2.1. Cell Lines

Ten human melanoma cell lines (A375, A375M, CHL-1, MEL224, MEL501, MEL505, MEWO, RPMI7951, SKMEL24, and SKMEL3) were kindly provided by Andreja Ambriović Ristov, PhD and Neda Slade, PhD. Cell lines HS895.SK (ATCC CRL-7636; Accession number CVCL_0992), HS895.T (ATCC CRL-7637; Accession number CVCL_0993), HS940.T (ATCC CRL-7691; Accession number CVCL_1038) and SKMEL2 (ATCC HTB-68; Accession number CVCL_0069) were purchased from the ATCC (Manassas, VA, USA). HS895.SK cell line represents a healthy control: skin keratinocytes isolated from the same patient as the HS895.T melanoma cell line. All cell lines were maintained in recommended medium: Dulbecco’s Modified Eagle Medium (Merck KgaA, Darmstadt, Germany), RPMI 1640 medium (Merck KgaA, Darmstadt, Germany), or Eagle’s Minimum Essential Medium (Merck KgaA, Darmstadt, Germany), supplemented with 10% FBS (Merck KGaA, Darmstadt, Germany), 1 mM sodium pyruvate, 1% streptomycin/penicillin and 4 mM L-glutamine (Gibco Thermo Fisher Scientific, Waltham, MA, USA).

### 2.2. Plasmids and Cell Transfection

For the *GLI* transfection experiments, cells were plated at density of 3 × 10^5^ cells/well in 6-well plates, left to attach for 24 h, and then transfected with 5 μg of *GLI* expression plasmids: GLI1 (pcDNA4NLSMTGLI1, kindly gifted by Fritz Aberger, PhD), GLI2 and GLI3 (p4TO6MTGLI2, pcDNA4/TO/GLI3richtig, both a kind gift from Milena Stevanović, PhD) using the X-fect reagent (Clontech, Mountain View, CA, USA) following the manufacturer’s instructions.

### 2.3. MTT Assay

For determining cell viability and proliferation, compound 3-(4,5-Dimethylthiazol-2-yl)-2,5-diphenyltetrazolium bromide (MTT) was used as previously described [20]. Cells were treated with different HH-GLI pathway inhibitors using the concentration ranges that correspond to previously published studies: GANT61 5–25 μM (Selleck Chemicals, Houston, TX, USA), cyclopamine (CYC) 1.25–10 µM (Selleck Chemicals, Houston, TX, USA), and lithium chloride (LiCl) 5–40 mM (Kemika, Zagreb, Croatia) for 24–72 h [21,22,23]. Cell viability was measured on LabsSystems Multiskan MS microplate reader (Thermo Fisher Scientific, Waltham, MA, USA) at 570 nm. The treatment was performed in quadruplicate for each dose, and the experiment was repeated twice. 

### 2.4. Western Blot

Whole-cell protein extraction, determining the protein concentration, and western blot technique were performed as previously described [20]. The membranes were probed with following primary antibodies: rabbit anti-GLI1 1:300 (V812, Cell Signaling Technology, Danvers, MA, USA), mouse anti-GLI2 1:100 (sc-271786, Santa Cruz Biotechnology, Dallas, TX, USA), rabbit anti-GLI3 1:1000 (GTX104362, GeneTex, Irvine, CA, USA), rabbit anti-PTCH1 1:1000 (17520-1-AP, ProteinTech, Rosemont, IL, USA) and mouse anti-β-actin 1:4000 (60008-1-Ig, ProteinTech, Rosemont, IL, USA) was used as loading control. After overnight incubation, membranes were washed in TBST (Tris-Buffered Saline, 0.1% Tween^®^ 20 Detergent) and incubated for 1 h with appropriate secondary HRP-conjugated antibodies, anti-rabbit 1:6000 (554021, BD Pharmingen, San Jose, CA, USA) and anti-mouse 1:8000 (554002, BD Pharmingen, San Jose, CA, USA). Proteins were visualized using SuperWest Signal Pico and Femto reagents (Thermo Fisher Scientific, Waltham, MA, USA) on Uvitec Image Alliance 4.7 instrument (UVItec, Cambridge, England, UK).

### 2.5. RNA-Sequencing

Human melanoma cell lines CHL-1 (wild-type for both NRAS and BRAF), MEL224 (NRAS^Q61R^), and A375 (BRAF^V600E^) were transfected with expression plasmids for *GLI1*, *GLI2* or *GLI3* in two independent experiments. Non-transfected lines were used as controls. Briefly, 200,000 cells were seeded in a Ø10 cm cell culture dish and transfected the next day with 5 µg of plasmid DNA, using the X-fect reagent (Clontech, Mountain View, CA, USA) and following the manufacturer’s instructions. Forty-eight hours post-transfection, total RNA was isolated with Absolutely RNA miRNA Kit (Agilent Technologies, Santa Clara, CA, USA). The quality of the isolated RNA was checked on agarose gel, and RNA concentrations and purity were measured on NanoPhotometer N60 instrument (Implen, Munich, Germany). Sequencing libraries were generated and sequenced by DNA Link Company (Seoul, South Korea). The integrity of RNA samples was checked on Agilent 2100 Bioanalyzer (Agilent Technologies, Santa Clara, CA, USA). The instrument provided data on RNA concentration, electropherogram, rRNA subunit ratios, and RNA integrity number (RIN). All samples had RIN ≥ 8.0 and a 28S:18S ratio ≥ 1.4. Total amount of 1 µg of RNA was used. cDNA libraries were prepared using TruSeq mRNA library Kit, and sequencing was performed on Novaseq 6000 instrument (Illumina, San Diego, CA, USA). 

### 2.6. Chromatin Immunoprecipitation Sequencing (ChIP-seq)

Chromatin immunoprecipitation was performed using SimpleChIP Plus Enzymatic Chromatin IP kit (Cat. no. #9005, Cell Signaling Technology, Danvers, MA, USA) following the manufacturer’s protocol. Three cell lines with the strongest basal expression of GLI proteins were selected for the experiment (CHL-1, A375, and MEL224). Cells were cultured until 80–90% confluence in 150 cm^2^ flasks. To preserve protein–DNA interactions, cells were fixed twice, first using 2 mM Di(N-succinimidyl) glutarate (DSG) (Merck KGaA, Darmstadt, Germany) followed by 1% formaldehyde. Samples were incubated with specific ChIP-grade antibodies against GLI1 10 μg/sample (AF3324, R&D Systems, Minneapolis, MN, USA) [24], GLI2 4 μg/sample (AF3526, R&D Systems) [24], and GLI3 10 μg/sample (AF3690, R&D Systems) overnight. Library preparation for next-generation sequencing was performed using SimpleChip ChIP-seq DNA Library Prep Kit for Illumina (Cat. no. #56795, Cell Signaling Technology) and SimpleChIP ChIP-seq Multiplex Oligos for Illumina (Dual Index Primers) (Cat. no. #47538, Cell Signaling Technology) according to manufacturer’s protocol. Libraries were prepared for 24 samples which included selected three cell lines for each GLI transcription factor with INPUT control in biological duplicates. A total of 5 ng of purified chromatin was used for each library preparation, followed by adaptor ligation. Samples were then purified from unbound adaptors using magnetic AMPure XP beads (Beckman Coulter, Brea, CA, USA), eluted, and amplified with a unique combination of dual index primers. Amplified chromatin fragments were additionally purified using magnetic beads. Quality control was performed on Bioanalyzer 2100 (Agilent Technologies, Santa Clara, CA, USA) and Qubit V1 (Invitrogen Thermo Fisher Scientific, Waltham, MA, USA). Next-generation sequencing was completed on NextSeq 500 (Illumina, San Diego, CA, USA).

### 2.7. Bioinformatic Analysis of RNA-seq Data

FastQC software (v.0.11.5) was used to assess the sequencing quality of the raw fastq data. The sequencing reads have been trimmed from adapters with Trim galore (v.0.3.7) and then mapped to human genome hg38 with STAR aligner (v.2.4.0d) with the following parameters: sjdbOverhang 99, outFilterMismatchNoverReadLmax 0.04, outFilterMultimapNmax 500, outSAMmultNmax 1. Mapped reads were then quantified with RSEM (v.1.2.26) over hg38 gencode transcriptome (v.28). Further analysis was performed with R (v.4.0.1), and differential gene expression was performed with edgeR (v.3.30.3) for each cell line with overexpressed GLI against the control. Gene Set Enrichment Analysis of KEGG pathways was performed with function gseKEGG from R package clusterProfiler (v.3.0.4). Data were visualized in R with custom scripts, available at https://github.com/NenBarto/GLI (accessed on 10 August 2022).

### 2.8. Bioinformatic Analysis of ChIP-seq Data

ChIP-seq libraries were first trimmed with trimgalore from FastQC (v.0.11.5) and mapped with BWA v0.7.9a to human genome hg38. After removing duplicates with picard-tools (v. 1.138), peaks were called with MACS (v.2.0.10) with parameters -f BAMPE -g -B -q 0.01. Further analysis was performed in R (v.4.0.1) and packages ChIPpeakAnno and ChIPseeker using custom scripts available at https://github.com/NenBarto/GLI (accessed on 10 August 2022). Motif enrichment was performed with MEME Suite v.5.3.0 [25]. The lists of differentially expressed genes (DEGs) identified by RNA-seq (FDR < 0.01, logFC > 1) were compared with the ChIP-seq identified lists of targets for each of the GLI proteins using the Venny 2.1 tool [26] and identified overlapping targets were selected for qPCR validation. DEGs for each of the GLI proteins were analyzed using the GeneAnalytics platform [27] to examine their involvement in different pathways and diseases. The targets that showed high fold change values, association with a large number of cancers, and involvement in the MAPK signaling pathway were selected for qPCR validation. 

### 2.9. Quantitative PCR Analysis

Total RNA was extracted using TRIzol reagent (Invitrogen, Carlsbad, CA, USA) following the manufacturer’s instructions. cDNA was generated from 1 μg of RNA using the High-Capacity cDNA synthesis kit (Thermo Fisher Scientific, Waltham, MA, USA) and qRT-PCR performed on CFX-96 instrument (Bio-Rad Laboratories, Hercules, CA, USA) using SsoAdvanced SYBR Green Supermix (Bio-Rad Laboratories, Hercules, CA, USA) with gene-specific primers. Fold change was calculated relative to the *RPLP0* and *TBP* housekeeping genes. Primer sequences used for qPCR are listed in Appendix A.

### 2.10. Statistical Analysis

D’Agostino–Pearson test was used for testing normality of data distribution. Non-normal data were log-transformed. An independent samples T-test or One-way ANOVA with Dunnett’s post hoc test was used for inferring the differences in gene expression. Statistical analyses were performed with MedCalc v19.2.1 (MedCalc Software bv, Ostend, Belgium). Two-tail *p*-values < 0.05 were considered statistically significant.

## 3. Results

### 3.1. HH-GLI Signaling Pathway in Melanoma Cell Lines with Different Genetic Background and Their Response to Pathway Inhibition

In order to confirm HH-GLI pathway activation in melanoma cell lines, we analyzed relative gene and protein expression levels of the pathway components (*GLI1*, *GLI2*, *GLI3,* and *PTCH1*) on a panel of 14 human melanoma cell lines with different genetic backgrounds. Five cell lines (HS 895.SK, MEWO, CHL-1, HS 895.T, and MEL501) are wild-type for both BRAF and NRAS gene; five cell lines (RPMI7951, SKMEL24, SKMEL3, A375M, and A375) have a BRAF^V600E^ mutation; three cell lines (HS 940.T, SKMEL2, and MEL224) have an NRAS^Q61R^ mutation, and the MEL505 cell line has a KRAS^G12V^ mutation. Figure 1A shows that PTCH1, which is considered the direct target of HH-GLI signaling, is detected in all cell lines. On the other hand, protein levels of GLI1, GLI2, and GLI3 are not detected equally in all melanoma cell lines. The highest expression levels of all three GLI proteins are detected in CHL-1, MEL501, A375, and MEL224 cell lines. GLI3 shows the most consistent expression in all cell lines in the full-length form (GLI3FL), while in some cell lines, it can also be found in the repressor form (GLI3R). Other authors have also detected the expression of GLI proteins in some of the melanoma cell lines we used in our study (MEWO, SKMEL2, MEL501, SKMEL3, and RMPI-7951) [28]. *BRAF* or *NRAS* mutation status is not correlated with the differences in protein or gene expression between the cell lines, but KRAS mutated MEL505 cell line shows lower gene expression levels for all tested genes (Appendix A). Out of the three *GLI* genes, *GLI3* shows the highest average gene expression among all groups, regardless of the mutational status (Figure 1B). Interestingly, cell line HS895.SK, which represents a healthy skin fibroblast control, showed no significant differences in gene expression levels compared to other melanoma cell lines, yet none of the analyzed proteins could be detected in this cell line. To test our hypothesis that in melanoma, activation of the HH-GLI signaling pathway is non-canonical due to its crosstalk with other signaling pathways, such as the MAPK pathway, we investigated how three HH-GLI pathway inhibitors (GANT61, CYC, and LiCl) affect cell viability and proliferation on 14 melanoma cell lines with different *BRAF* or *NRAS* mutation status (Figure 1C). Although we expected that cell lines with the highest expression of GLI proteins would be affected by the inhibition, this was not the case in our study. MTT assay showed that out of three HH-GLI inhibitors, the most effective is GANT61, a known direct GLI protein inhibitor. Cyclopamine, as an inhibitor of SMO, a membrane component of the canonical HH-GLI pathway activation, seems to have no or very little effect on the viability of melanoma cells, regardless of the dose increase or duration of the treatment (Figure 1C). We noticed that BRAF^V600E^ mutated cell lines seem to be more sensitive to GANT61 than cell lines that are wild-type for these genes, but in our case, this difference is not statistically significant (Appendix A). Additionally, we noticed a trend for higher sensitivity of metastatic cell lines to GANT61 compared to primary tumor cell lines, but again, the results are not statistically significant (Appendix A). In conclusion, the response to GANT61, a GLI inhibitor, compared to the poor response to cyclopamine, a SMO inhibitor, supports non-canonical HH-GLI pathway activation in melanoma cell lines.

### 3.2. RNA Sequencing Reveals Unique and Overlapping Targets of GLI Transcription Factors

As the first step in identifying novel GLI transcriptional targets that are in crosstalk with MAPK or other signaling pathways dysregulated in melanoma and that could be considered for combination therapy, we performed RNA sequencing on melanoma cell lines with overexpressed GLI1, GLI2, and GLI3 (Figure 2A and Appendix A show MDS plots). To our knowledge, there is no extensive data on the targets of all three GLI proteins in melanoma and their overlap. One study showed GLI1 and GLI2 transcriptional targets in primary neoplastic chondrocytes, detected by ChIP-seq and microarray methods, but in this study, only three targets were validated by qPCR [24]. Thus far, no one has taken into consideration transcriptional targets of all three GLI proteins. Across all cell lines, we found 1642 targets that were overlapping for GLI1 and GLI2, 23 overlapping targets of GLI2 and GLI3, and only 9 overlapping targets of GLI1 and GLI3. In total, we found 150 GLI1, GLI2, and GLI3 overlapping targets (Figure 2B). There were 607 unique targets of GLI1, 1080 unique targets of GLI2, and 37 unique targets of GLI3. After filtering according to FDR < 0.01, we identified a total of 808 DEGs (631 upregulated, 183 downregulated) for GLI1. For GLI2, we found 941 DEGs (711 upregulated, 230 downregulated). For GLI3, there were only 58 DEGs (35 upregulated and 23 downregulated) found using this method (Appendix A). Top scoring DEGs across all three cell lines are shown in Figure 2C. To identify pathways that are significantly represented in our list of differentially expressed genes, we performed pathway enrichment analysis. Figure 2D shows that some of the most enriched pathways in the case of GLI1 and GLI2 overexpression are Wnt signaling pathway, MAPK signaling pathway, and Ras signaling pathway. About 20–30% of DEGs are involved in these signaling pathways. An even bigger percentage of genes (30–40%) show involvement in Neuroactive ligand−receptor interactions. There is also significant involvement of DEGs in different cancer types (Figure 2D, Appendix A). Our findings confirm the crosstalk of HH-GLI with other signaling pathways, and to our knowledge, this is the first study that considered transcriptional targets of all three GLI proteins in melanoma.

### 3.3. Chromatin Immunoprecipitation (ChIP) Sequencing Reveals Novel Binding Targets of GLI Proteins

ChIP sequencing was used to identify GLI1, GLI2, and GLI3 binding regions in human melanoma cell lines and to further confirm RNA-seq results. For this purpose, cell lines with the highest endogenous GLI protein expression levels (CHL-1, A375, and MEL224) were selected. ChIP-seq datasets were merged across cell lines to observe overall GLI binding sites and increase the signal-to-noise ratio. More than 80% of ChIP-seq peaks were identified in the intergenic regions (Figure 3A,B), which corresponds to their previously observed enhancer binding properties [29]. Overall, the three TFs shared most of the sites, with GLI2 containing the largest number of unique binding loci (Figure 3C). We identified 2183 genes that contained GLI TFs binding sites in their promoters: 527 for GLI1 (24%), 1103 for GLI2 (50%), and 553 for GLI3 (25%) (Figure 3C). As expected, only a small proportion of genes had all three GLI TFs in their promoter regions (157 out of 2183, 7.2%), with a much larger proportion of GLI3-specific promoter binding (35.8%) (Appendix A). The binding sites for GLI TFs were centered around the gene transcription start sites (Figure 3D) and were enriched for previously established motifs for GLI1 (MA1990.1), GLI2 (MA0734.1), and GLI3 (MA1491.1), with *p*-values of 4.71 × 10^−61^, 8.4 × 10^−7^ and 1.64 × 10^−75^, respectively. *PTCH1,* as a known target, showed two peaks in the transcriptional start site (TSS), which was previously reported for GLI2 [30], and its position corresponds to the H3K4me3 region. One of the newly identified target genes, *EBI3*, shows a broad peak in the promoter region of the gene (Figure 3E).

### 3.4. qPCR Validation of RNA Sequencing and ChIP Sequencing Data Confirms 15 Novel GLI Target Genes

To validate the biological reproducibility of the results of DEG analysis, we performed qPCR experiments on seven melanoma cell lines. Cell lines were chosen to represent the different mutational backgrounds: A375 (BRAF^V600E^ homozygous), SKMEL24 (BRAF^V600E^ heterozygous), MEL224 (NRAS^Q61R^ homozygous), SKMEL2 (NRAS^Q61R^ heterozygous), MEL505 (KRAS^G12V^ heterozygous), CHL-1 and MEWO (wild-type for BRAF and NRAS). Selected cell lines were transfected with *GLI1*, *GLI2,* or *GLI3* expression plasmids, and gene expression of a total of 23 genes (21 novel targets, plus *PTCH1* and *GLI1* as known targets) was determined. To narrow down a list of targets for qPCR validation, two approaches were used (Figure 4A). The first approach identified potential targets under direct transcriptional control of GLI proteins by comparing the list of DEGs obtained by RNA-seq with the list of genes identified by ChIP-seq analysis for all three cell lines. For GLI1, 808 DEGs were compared with 231 identified ChIP-seq GLI1 targets, identifying three common targets, namely: *FLG*, *SAMMSON*, and *SPRY2*. For GLI2, 941 DEGs were compared with 470 ChIP-seq GLI2 targets, identifying 11 common targets: *HES1, EBI3, CACNA2D2*, *LAPTM5*, *LY6D*, *FLG*, *GLI1*, *PTCH1*, *RDH10*, *STK32C*, and *RAB34*. Among the identified targets, there were two known HH-GLI pathway targets (*PTCH1* and *GLI1*), confirming the validity of the selection process. A comparison of GLI3 RNA-seq and ChIP-seq data showed no common targets. The chIP-seq analysis identified GLI1 or GLI2 binding motifs in all 11 identified targets, suggesting they are direct transcriptional targets of GLI1 and/or GLI2 proteins. The second approach was to analyze the DEGs from RNA-seq independently of the ChIP-seq data to identify potential indirect targets. After filtering by FDR and logFC values, the GeneAnalytics tool of the GeneCards database (genecards.org, RRID:SCR_002773) was used to identify the role of DEGs in signaling pathways and diseases. Several categories of pathways and diseases with a high relevance score were considered when selecting genes of interest: “Pathways in cancer”, “PI3K-AKT signaling pathway”, “MAPK signaling pathway”, “WNT/Hedgehog/NOTCH”, “neoplasm”, “melanoma” and “abnormalities of the skin”, as well as logFC values of these genes in RNA-seq analysis. Additional screening was performed based on their expression in melanoma from the GEPIA database [31] (SKCM dataset vs TCGA normal and GTEx data, N(T) = 461, N(N) = 558) and The Human Protein Atlas (proteinatlas.org) [32] (RNA expression and staining of melanoma) as well as survival data from GEPIA database. With this approach, we were able to identify ten targets: *KRT16*, *KRT17*, *S100A7*, *S100A9*, *GH1*, *SOX9*, *BIRC7*, *MRAS*, *RET*, and *IL1R2*. Finally, 21 targets were selected for qPCR validation: 10 identified by RNA-seq only (*KRT16*, *KRT17*, *S100A7*, *S100A9*, *GH1*, *SOX9*, *BIRC7*, *MRAS*, *RET* and *IL1R2*) and 11 targets identified with both ChIP-seq and RNA-seq (*HES1*, *FLG*, *RAB34*, *SAMMSON*, *SPRY2*, *CACNA2D2*, *LAPTM5*, *LY6D*, *RDH10*, *STK32C* and *EBI3*) (Figure 4B,C). A summary of known functions of these targets and their role in cancer is shown in Table 1. Identified targets, including *KRT16, KRT17, S100A7, MRAS, BIRC7, IL1R2,* as well as several direct GLI targets (confirmed by both RNA-seq and ChIP-seq)—*RAB34*, *LAPTM5*, *RDH10*, and *STK32C—*exhibited a consistent and uniform expression pattern. In addition, the genes *EBI3*, *GH1*, *SOX9*, *RET*, and *SPRY2* are also good candidates for HH-GLI targets but exhibited a less uniform expression pattern in these melanoma cell lines. Overall, by combining RNA-seq and ChIP-seq results and elaborate filtering of these genes, we successfully validated 15 novel targets of GLI proteins in melanoma cell lines.

## 4. Discussion

BRAF inhibitors have improved patient survival compared with standard chemotherapy, but these benefits are not persistent, as most patients develop resistance to therapy which leads to disease progression [103]. There are different mechanisms that can activate a variety of signaling pathways, thereby bypassing the effect of BRAF inhibition. It is already known that the HH-GLI signaling pathway is active in melanoma [11,12,28,104]. Here, we confirm HH-GLI pathway activity in 14 melanoma cell lines with different genetic backgrounds. Several studies also show that inhibition of the HH-GLI pathway can decrease melanoma cell proliferation [11,12,105,106,107]. The HH-GLI pathway inhibitors affect signal transduction at different levels. Cyclopamine inhibits the SMO protein on the cell membrane [108]. In contrast, lithium chloride increases the phosphorylation of Ser9 residue on GSK3β kinase, which regulates GLI protein activity at the post-translational level. GSK3β phosphorylates GLI3 and thereby promotes its processing into GLI3R, which downregulates the HH-GLI pathway [109]. ATO and GANT61, in turn, affect the activity of GLI proteins [110,111]. One of these studies pointed out that primary melanoma cell lines with *BRAF* mutation are more sensitive to SMO inhibitor, sonidegib, than *BRAF* wild-type cells [11]. We also noticed that BRAF^V600E^ mutated cell lines seem to be more sensitive to inhibitor GANT61 than cell lines that are wild-type for these genes, but in our case, this difference is not statistically significant. Thus far, studies have demonstrated that in colon cancer, neuroblastoma, and pancreatic cancer, GANT61 is the most effective inhibitor of cell growth among all tested HH-GLI inhibitors [112,113,114]. On the other hand, one study in melanoma shows comparable effects of GANT61 and cyclopamine [115]. Our MTT-assay results show that cyclopamine seems to have no or very little effect on the viability of melanoma cell lines, while the most effective inhibitor in melanoma cell lines was, indeed, GANT61. This result also supports the assumption that, in the case of melanoma, non-canonical pathway activation is likely more important than canonical [3,19,104,107]. 

Because the exact interplay between the HH-GLI pathway and MAPK signaling pathway is not yet understood, we decided to investigate the transcriptional targets of all three GLI proteins in melanoma with different genetic backgrounds, either harboring a *BRAF* mutation, an *NRAS* mutation, or no mutation in these two genes. We applied RNA sequencing and combined it with ChIP sequencing to identify direct but also unique and overlapping targets of GLI1, GLI2, and GLI3 in three melanoma cell lines. By doing so, we identified a total of 808 DEGs for GLI1, 941 DEGs for GLI2, and 58 DEGs for GLI3. KEGG analysis confirmed that many of the identified DEGs are involved in various signaling pathways, including MAPK, Ras, Hippo, and Wnt pathways [3,116], as well as in many types of cancer [8,117,118,119]. Our next step of carefully screening and filtering targets led us to select 21 targets for qPCR validation. We successfully validated 15 novel targets of GLI proteins that were not identified in any previous study. To our knowledge, this is the first comprehensive study of transcriptional targets of all three GLI proteins in melanoma. Identified targets, such as *KRT16*, *KRT17*, *S100A7*, *MRAS*, *BIRC7*, *IL1R2*, as well as several direct GLI targets (confirmed by both RNA-seq and ChIP-seq)—*RAB34*, *LAPTM5*, *RDH10,* and *STK32C*—have a consistent and uniform expression pattern. Expression levels of these genes are increased with GLI1 or GLI2 overexpression, regardless of the mutational status of the cell lines. We were also able to validate genes *EBI3*, *GH1*, *SOX9*, *RET,* and *SPRY2* as HH-GLI targets. Their expression was consistent in the majority of the cell lines, with few exceptions. Six targets—*S100A9*, *FLG*, *SAMMSON*, *LY6D*, *CACNA2D2*, and *HES1*—were found to have variable expression in different melanoma cell lines, so they could not be validated as GLI targets in melanoma. 

Table 1 represents a summary of protein functions and already published roles in cancer for 21 discovered GLI targets. Thus far, 8 out of 21 GLI targets we chose to validate—*KRT16*, *S100A9*, *SOX9*, *BIRC7*, *EBI3*, *FLG*, *SAMMSON* and *SPRY2*—were already implicated in melanoma pathogenesis [34,46,47,51,52,64,79,81,82,87,101,120]. KRT16, a regulator of innate immunity in the skin, seemed to be significantly downregulated in metastatic melanoma and was also found to be the highest discriminator between prognostic and metastatic melanoma [34]. Our RNA-seq and qPCR results show that out of all targets, KRT16 is by far the most upregulated overlapping target of GLI1 and GLI2, with a logFC value of 12.5 obtained by RNA-seq and log2FC value that goes up to 16 in qPCR experiments, depending on the cell line. LogFC values and expression patterns of *KRT17* closely follow those of *KRT16* in all seven melanoma cell lines (Figure 4C). 

*S100A9* is suggested to have a role in acquired resistance to BRAF inhibitors [47] and in melanoma metastasis [46]. Our results show that *S100A9* is expressed only in one tested melanoma cell line, A375 (BRAF^V600E^ mut). In the majority of our melanoma cell lines, S100A9 expression could not be detected. By contrast, *S100A7* shows a much wider expression pattern than *S100A9*, but despite its ability to promote cell proliferation, migration, invasion, and tumor metastasis in cervical, breast, and ovarian cancer [42,43,44,45], S100A7 has not previously been implicated in melanoma pathogenesis. From other validated targets in this study that have not yet been investigated in melanoma, we would like to point out *MRAS*, *IL1R2*, *RAB34*, *LAPTM5*, *RDH10*, and *STK32C*. Similar to *S100A7*, we considered them important because of their involvement in processes of tumor cell proliferation, migration, and invasion, or their interactions with members of the MAPK cascade. For example, it is shown that cells overexpressing MRAS have higher migratory potential and that MRAS/SHOC2/SCRIB complex coordinates ERK pathway dynamics [56]. It has been proposed that increased IL1R2 levels are important during the initiation and progression of human gastric cancer [70]. One study demonstrates the existence of a novel mechanism of tyrosine phosphorylation of RAB34 in regulating cell migration, invasion, and adhesion through modulating the endocytosis, stability, and recycling of integrin β3 [84]. It has been shown that inhibition of LAPTM5 blocks bladder cancer cell proliferation and cell cycle via deactivation of ERK1/2 and p38 [121] and that silencing of *STK32C* inhibited tumor cell proliferation, migration, and invasion in human bladder cancer cells [99]. Finally, RDH10 overexpression has an antiproliferative effect on hepatocellular carcinoma cell lines [98].

Although we identified *FLG* as the overlapping target of GLI1, GLI2, and GLI3 and detected it with both ChIP-seq and RNA-seq, its expression levels detected by qPCR experiments are not consistent between the cell lines. A375 cell line shows increased expression levels of *FLG* in all GLI overexpressed samples, while SKMEL24 shows a decrease in *FLG* expression levels. In the other cell lines, *FLG* expression could not be detected with qPCR. A previous study has noted the important role of the long noncoding RNA (lncRNA) *SAMMSON* in melanoma [86]. More recently, *SAMMSON* has been shown to be important for human melanoma cell growth and survival while also highlighting the role of a *SAMMSON* in modulating the adaptive resistance of mutant *BRAF* melanoma to RAF inhibitors [87]. Similar to *SAMMSON, SPRY2* has also been implicated in resistance to BRAF inhibitors [101,120]. Our RNA-seq results reveal that both *SAMMSON* and *SPRY2* are downregulated targets of GLI1, and ChIP-seq confirms that *SAMMSON* has a GLI1 binding motif, while *SPRY2* contains GLI1 and GLI2 binding motifs. Because GLI1 is a transcriptional activator, it is not clear how it downregulates the expression of these two targets. It is likely that some other factors, apart from GLI1, play a role in their regulation. qPCR results show that *SAMMSON* expression levels vary among the cell lines. For example, A375 and SKMEL2 (with GLI1 or GLI2 overexpression) exhibit decreased expression levels of *SAMMSON*, while other cell lines, such as SKMEL24, MEL224, and MEL505, generally show increased *SAMMSON* expression levels. qPCR results show that *SPRY2* is detected in all melanoma cell lines, especially those with GLI3 overexpression.

## 5. Conclusions

Our studies confirm that in melanoma, the HH-GLI signaling pathway is in crosstalk with other signaling pathways and that its activation is more likely non-canonical than canonical. Out of 21 selected targets, we validated 15 as novel targets of GLI proteins, considering their expression in melanoma cell lines and possession of GLI binding motifs. Our study provides insight into the unique and overlapping transcriptional output of the GLI proteins in melanoma, which will contribute to a better understanding of the GLI code and its role in tumorigenesis. Other potential targets can also be functionally validated using this data in the future, especially by researchers in the HH-GLI field that are interested in other aspects of HH-GLI signaling. Our findings provide new potential targets to consider while designing melanoma-targeted therapy, especially in the case of recurrent disease due to therapy resistance.

## Figures and Tables

**Figure 1 cancers-14-04540-f001:**
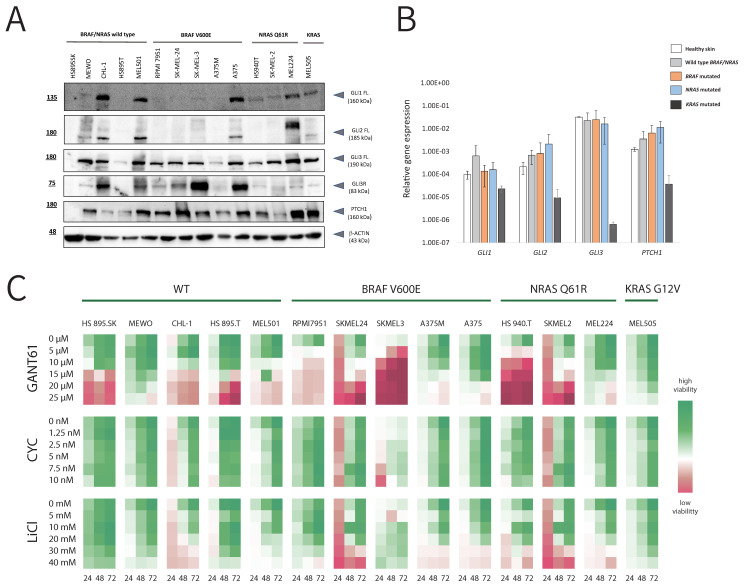
HH-GLI pathway activity in melanoma cell lines. (**A**) Western blot analysis of relative protein expression levels of GLI1, GLI2, GLI3 and PTCH1 in a panel of 14 melanoma cell lines. FL refers to the full-length protein, while R refers to the repressor form. (**B**) Average gene expression of *GLI1*, *GLI2*, *GLI3* and *PTCH1* relative to the housekeeping gene *RPLP0* summarized according to the mutational background of the tested panel of melanoma cell lines. (**C**) Heatmap showing MTT proliferation assay on 14 melanoma cell lines. Cells were treated with three different HH-GLI pathway inhibitors in five doses, during 24, 48 and 72 h. Green color indicates high cell viability and red color indicates low viability (cell death). The uncropped blots are shown in Appendix A.

**Figure 2 cancers-14-04540-f002:**
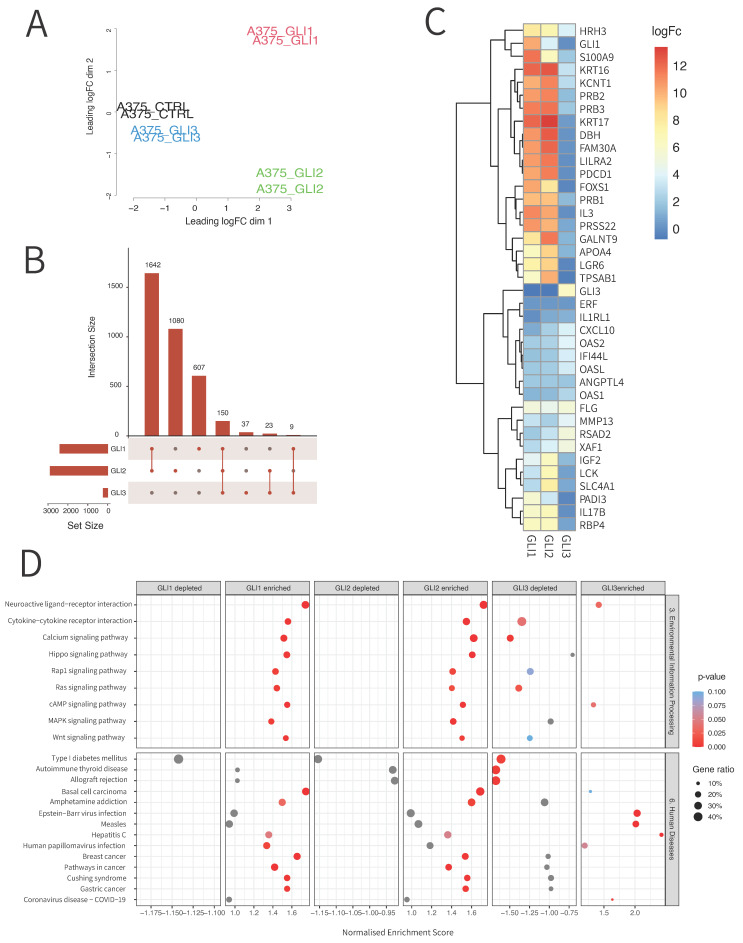
Expression profiles of CHL-1, A375, and MEL224 cell lines with transfected *GLI1*, *GLI2* and *GLI3*. (**A**) MDS plot of A375 cell line. (**B**) Upset plot of differentially expressed genes across all cell lines and GLI proteins. (**C**) Heatmap of expression of top DEGs sorted by logFC across all cell lines. (**D**) Gene enrichment analysis of DEGs across cell lines, transfection with GLI1-3 vs. control. Upper half of the plot shows KEGG pathway analysis, while the lower part shows categories of diseases. On *x*-axis: normalized enrichment scores. Size of the circles denote ratio of DE genes in pathways. Color denotes significance, with gray and blue circles denoting non-significant enrichments and red denotes significant enrichment.

**Figure 3 cancers-14-04540-f003:**
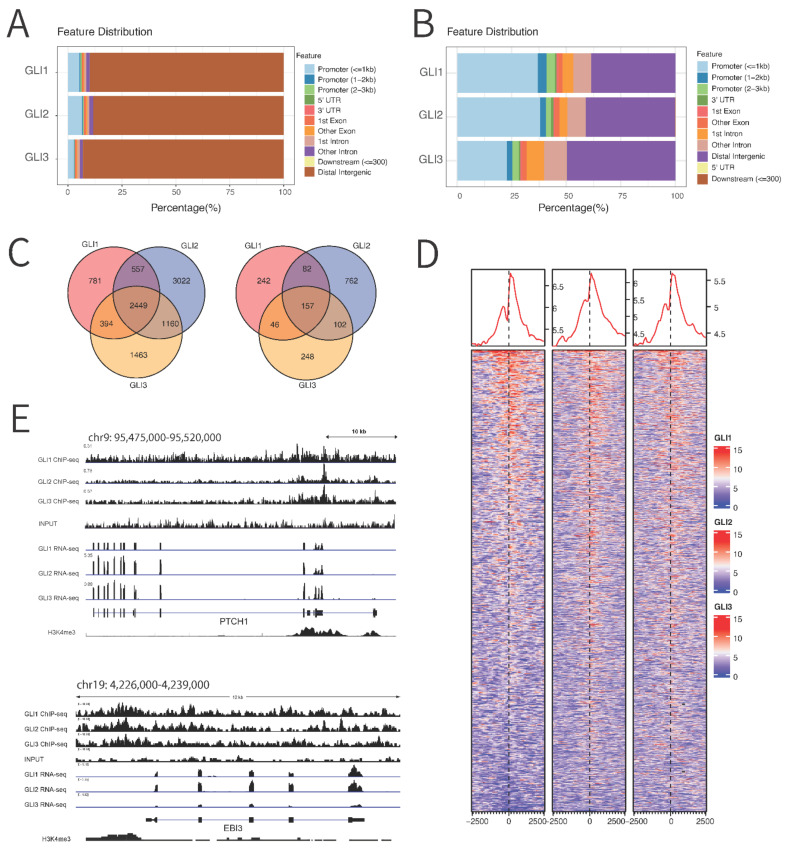
ChIP-seq analysis of GLI1-3 binding across cell lines. (**A**). Binding of peaks on the whole genome. (**B**) Binding of peaks that centered on the promoter regions (**C**) Overlap of GLI1-3 binding sites: left image shows overlapping of peaks on whole genome, and right image shows overlapping of peaks on the promotor regions only. FDR value vas set to 0.2. (**D**) Heatmap of binding of GLI1-3 TFs across promoter sites. (**E**) Peak distribution in *PTCH1* and *EBI3* loci.

**Figure 4 cancers-14-04540-f004:**
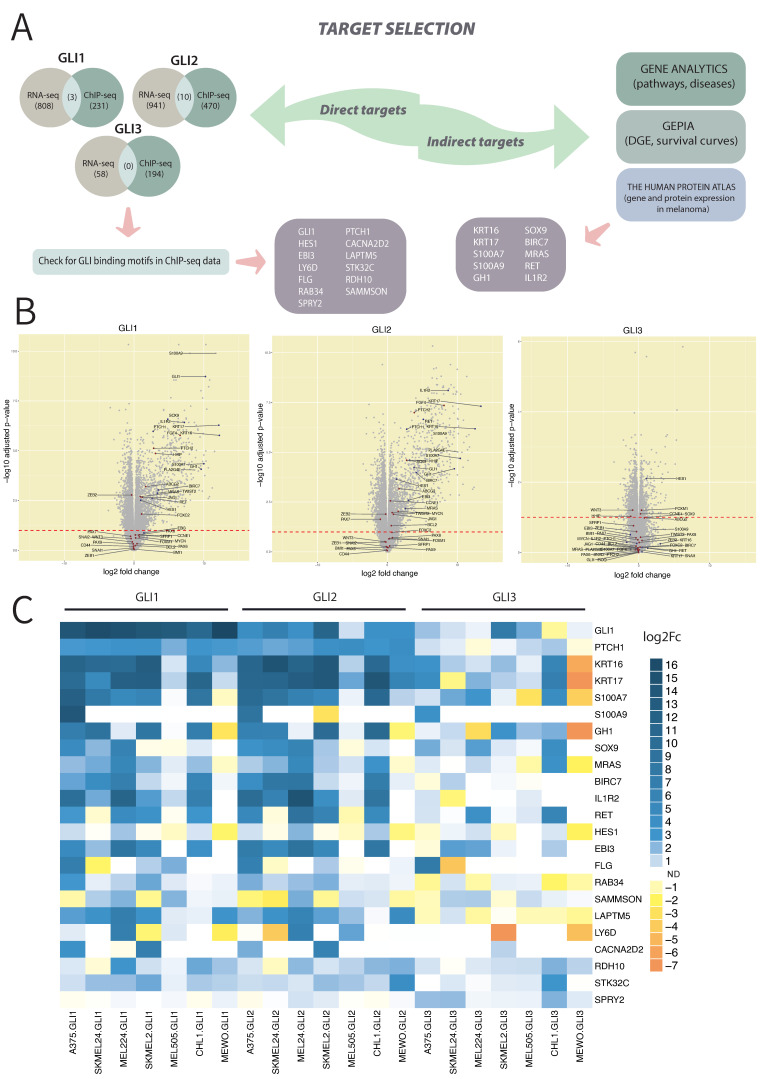
Validation of 21 selected DEGs. (**A**) Schematic representation of choosing DEGs for qPCR validation. (**B**) Volcano plot represents previously identified targets of HH-GLI signaling in red and targets selected for validation in this study in blue for each GLI protein. (**C**) Heatmap showing qPCR validation of GLI target genes identified by both ChIP-seq and RNA-seq on seven melanoma cell lines (A375, SKMEL24, MEL224, SKMEL2, MEL505, CHL-1 and MEWO) with overexpressed GLI1, GLI2 or GLI3. The experiment was repeated two times in triplicates. Validation was performed for 21 DEGs and two known pathway targets *PTCH1* and *GLI1* as controls. The relative expression level of each gene was determined using the 2^−ΔΔCt^ method with *RPLP0* as the internal reference gene. Heatmap shows log2 fold change values, ND stating that expression levels could not be detected after 37th cycle.

**Table 1 cancers-14-04540-t001:** Summary of known functions and roles of 21 selected GLI target genes in cancer.

Gene	Function According to Gene Cards	Role in Cancer	Reference
*KRT16*	type I keratin that regulates innate immunity in response to skin barrier breach	regulates immune response, metastasis, cancer stemness and drug resistance in melanoma, SCC, and breast cancer	[33,34,35,36]
*KRT17*	type I keratin involved in regulation of protein synthesis and epithelial cell growth	regulates therapy resistance, proliferation, migration and invasion in CRC, pancreatic cancer, and NSCLC	[37,38,39,40]
*S100A7*	member of the S100 family of proteins involved in the regulation of cell cycle and differentiation	regulates tumor invasion, angiogenesis, migration, EMT and chemoresistance in melanoma, cervical cancer, and ovarian cancer	[41,42,43,44,45]
*S100A9*	calcium- and zinc-binding protein involved in immune response	regulates chemoresistance, cell invasion and metastasis in melanoma, cervical carcinoma, and prostate cancer	[46,47,48,49]
*GH1*	member of the somatotropin/prolactin family of hormones important for growth control	dysregulates MAPK pathway and blocks cell motility in colon cancer	[50]
*SOX9*	transcription factor important for differentiation and skeletal development	regulates metastasis, cell invasion, migration and stemness in melanoma, CRC, and esophageal cancer	[51,52,53,54]
*MRAS*	Ras GTPase that functions as signal transducer in cell growth and differentiation	regulates MAPK pathway and drives tumorigenesis in gastric cancer and prostate cancer	[55,56,57]
*BIRC7*	member of the inhibitor of apoptosis protein family associated with cancer progression and chemotherapy resistance	regulates chemoresistance and can serve as a biomarker in prostate cancer, melanoma, and lung cancer	[58,59,60,61,62,63,64,65,66]
*IL1R2*	cytokine receptor that belongs to the interleukin 1 receptor family	regulates proliferation, angiogenesis and tumorigenesis initiation in breast cancer, melanoma, gastric cancer, and CRC	[67,68,69,70,71]
*RET*	receptor tyrosine-protein kinase involved in cell proliferation, migration, and differentiation	RET fusion are associated with tumorigenesis of chronic myelomonocytic leukemia	[72]
*HES1*	transcriptional repressor involved in cell differentiation, cell cycle, apoptosis, and self-renewal	regulates cell proliferation, invasion and self-renewal in CRC, breast cancer and glioblastoma	[73,74,75,76]
*EBI3*	secretory glycoprotein belonging to the hematopoietin receptor family involved in IL-27 formation	EBI3 overexpression is associated with poor prognosis of breast and cervical cancer and impaired immune response in melanoma	[77,78,79,80]
*FLG*	intermediate filament-associated protein that aggregates keratin intermediate filaments in mammalian epidermis	regulates growth and angiogenesis and can be valuable in prognosis and treatment of melanoma	[81,82]
*RAB34*	small GTPase involved in protein transport and ciliogenesis pathways	regulates cell adhesion, migration and invasion in breast cancer and correlates with tumor progression of HCC and glioma	[83,84,85]
*SAMMSON*	lncRNA with crucial role in cell survival and mitochondrial metabolism	regulates therapy response and mitochondrial function in melanoma	[86,87]
*LAPTM5*	transmembrane receptor associated with lysosomes	potential biomarker for HCC, glioblastoma, and testicular cancer	[88,89,90]
*LY6D*	marker at earliest stage specification of lymphocytes between B-and T-cell development	therapy outcome and survival prediction in BCC, prostate cancer, NSCLC, laryngeal cancer, and breast cancer	[91,92,93,94,95]
*CACNA2D2*	alpha-2/delta subunit of the voltage-dependent calcium channel complex	regulates cell proliferation and angiogenesis in prostate cancer, while CACAN2D2 inhibition induces NSCLC tumorigenesis	[96,97]
*RDH10*	retinol dehydrogenase essential for organ development	RDH10 overexpression has an antiproliferative effect in hepatocellular carcinoma	[98]
*STK32C*	serine/threonine protein kinase	STK32C overexpression in bladder cancer contributes to tumor progression	[99]
*SPRY2*	inhibitor of RTK signaling proteins activity	SPRY2 inhibits cell growth and therapy resistance occurrence via MAPK pathway in melanoma and hepatocellular carcinoma	[100,101,102]

## Data Availability

Raw and processed RNA-seq and ChIP-seq sequencing data can be found in the ArrayExpress database under the following ID: E-MTAB-11936. Complete gene lists resulting from all analyses are provided as Appendix A.

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
