# Peer review of "RNA-seq and ChIP-seq Identification of Unique and Overlapping Targets of GLI Transcription Factors in Melanoma Cell Lines"

_cancers, 2022, doi:10.3390/cancers14184540_

Round 1

Reviewer 1 Report

In this manuscript, Kurtović and colleagues applied RNA sequencing combined with ChIP sequencing to identify direct, but also unique and overlapping targets of GLI1, GLI2 and GLI3 transcription factors, the final effectors of the Hedgehog pathway, in three melanoma cell lines with different genetic background (BRAF or NRAS mutation, and BRAF/NRAS wt). They identified a total of 808 differentially expressed genes (DEGs) for GLI1, 941 DEGs for GLI2 and 58 DEGs for GLI3. After further steps of screening and filtering, authors confirmed and validated 15 novel direct GLI targets in melanoma. The identified targets have a consistent and uniform expression pattern in melanoma cell lines and possess the GLI binding motif.

To my knowledge this is the first comprehensive study of transcriptional targets of all three GLI proteins in melanoma. The RNA-seq and ChIP-seq data presented in this manuscript represent a valuable source of potential targets that could be functionally validated by researchers working in the Hedgehog field. In addition, since some of the identified targets are already targetable, this study could provide new potential targets to consider while designing melanoma targeted therapy or immune therapy.

A few minor issues should be addressed:

1)    Fig. 1A-C: healthy skin fibroblasts (HS895.SK) showed no expression of GLI1, GLI2, GLI3 nor PTCH1 proteins, which makes to assume they do not have an active HH pathway. Why do they respond to treatment with the GLI inhibitor GANT61?

2)    Fig. 1C: more justification about the concentration of drugs (GANT61, Cyc, LiCl) being used.

3)    For readers who are not expert of HH signaling, I would mention in the text how LiCl represses the HH-GLI pathway (adding a reference).

4)    The differences in the number of targets identified by ChIP-seq could be ascribed to the fact that GLI2 is a better ChIP-grade antibody compared to those against GLI1 or GLI3?

5)    Supplementary Table 1 does not report the sequences of housekeeping genes (RPLP0 and TBP) used for qPCR.

Minor points:

-       Graph Fig. 1B: on the vertical axis replace “Relative gene espression” with “Relative gene expression”.

-       Line 51: a semicolon is missing after “repressors” to brake the sentence.

-       Line 128: correct “1 ug of RNA”.

Author Response

Reviewer 1

In this manuscript, Kurtović and colleagues applied RNA sequencing combined with ChIP sequencing to identify direct, but also unique and overlapping targets of GLI1, GLI2 and GLI3 transcription factors, the final effectors of the Hedgehog pathway, in three melanoma cell lines with different genetic background (BRAF or NRAS mutation, and BRAF/NRAS, wt). They identified a total of 808 differentially expressed genes (DEGs) for GLI1, 941 DEGs for GLI2 and 58 DEGs for GLI3. After further steps of screening and filtering, authors confirmed and validated 15 novel direct GLI targets in melanoma. The identified targets have a consistent and uniform expression pattern in melanoma cell lines and possess the GLI binding motif.

To my knowledge this is the first comprehensive study of transcriptional targets of all three GLI proteins in melanoma. The RNA-seq and ChIP-seq data presented in this manuscript represent a valuable source of potential targets that could be functionally validated by researchers working in the Hedgehog field. In addition, since some of the identified targets are already targetable, this study could provide new potential targets to consider while designing melanoma targeted therapy or immune therapy.

A few minor issues should be addressed:

1)    Fig. 1A-C: healthy skin fibroblasts (HS895.SK) showed no expression of GLI1, GLI2, GLI3 nor PTCH1 proteins, which makes to assume they do not have an active HH pathway. Why do they respond to treatment with the GLI inhibitor GANT61?

Answer: Thank you for your question. We also find this phenomenon interesting, but, unfortunately we don't have a clear answer to this question. In the literature, GANT61 is considered to be very specific inhibitor of GLI proteins. Even though HH-GLI pathway components were not detectable by Western blot, their gene expression was detected by qPCR, so this may be an issue of antibody sensitivity rather than an issue with the GANT61 inhibitor.

2)    Fig. 1C: more justification about the concentration of drugs (GANT61, Cyc, LiCl) being used.

Answer: We have tested doses that are below and above the calculated IC50 doses from the literature (5 different doses of each inhibitor). We have added the literature quotations into the Materials and Methods section to support this (lines 97-98, and the literature list has been updated). We are very grateful for your comment because we have noticed an error in the doses for cyclopamine, which is in micromolar range but was mistakenly written as nanomolar. This has now been double-checked and corrected in Materials and Methods and Figure 1.

3)    For readers who are not expert of HH signaling, I would mention in the text how LiCl represses the HH-GLI pathway (adding a reference).

Answer: Thank you for your comment and suggestion. We have inserted this paragraph in the Discussion, and included the mechanism for the other inhibitors as well, suported by references (lines 387-393, and the updated reference list).

4)    The differences in the number of targets identified by ChIP-seq could be ascribed to the fact that GLI2 is a better ChIP-grade antibody compared to those against GLI1 or GLI3?

Answer: That is an excellent comment. This may be the case, but we think it reflects the actual biological situation rather than antibody specificity because of the RNAseq data. Antibodies are crucial for the ChIP-sequencing, but no antibodies are used in RNA-sequencing, and the number of identified targets was also higher for GLI2 than for GLI1 and GLI3.

5)    Supplementary Table 1 does not report the sequences of housekeeping genes (RPLP0 and TBP) used for qPCR.

Answer: Thank you for noticing this, it was ommitted by mistake. We have now added RPLP0 and TBP sequences into the Supplementary table 1.

Minor points:

-       Graph Fig. 1B: on the vertical axis replace “Relative gene espression” with “Relative gene expression”.

-       Line 51: a semicolon is missing after “repressors” to brake the sentence.

-       Line 128: correct “1 ug of RNA”.

Thank you for the comments. We have made all the spelling and grammatical corrections, and a new Figure 1 has been uploaded.

Reviewer 2 Report

The study of Kurtovic et al. provides a comprehensive and novel analysis of isoform-specific GLI targets in melanoma cell lines with diverse mutational backgrounds that reveals common and unique direct and indirect targets. The study will be a valuable resource to other researchers in the Hedgehog field, and might guide identification of novel targets for melanoma therapy.

I would only like the authors to expand on the description of the mechanism of action -and molecular target - of LiCl for the non-Hh community, since it is not a specific Hh inhibitor.

Author Response

Reviewer 2

The study of Kurtovic et al. provides a comprehensive and novel analysis of isoform-specific GLI targets in melanoma cell lines with diverse mutational backgrounds that reveals common and unique direct and indirect targets. The study will be a valuable resource to other researchers in the Hedgehog field, and might guide identification of novel targets for melanoma therapy.

I would only like the authors to expand on the description of the mechanism of action -and molecular target - of LiCl for the non-Hh community, since it is not a specific Hh inhibitor.

Answer: Thank you for your comment and suggestion. We have inserted this paragraph in the Discussion part, and included the mechanism for the other inhibitors as well, suported by references (lines 387-393, and the updated reference list).

Reviewer 3 Report

The authors in this manuscript performed studies to identify new Hedgehog-GLI (HH-GLI)-regulated targets in melanoma and its crosstalk with MAPK pathway. For these studies, the authors utilized RNA sequencing and ChIP sequencing, and further validated their findings by qPCR. Presented data in the manuscript indicate that the protein levels of GLI1, GLI2 and GLI3 not detected equally in tested melanoma cell lines with different genetic background. The authors further utilized the melanoma cell lines that expressed elevated levels of all three GLI proteins and further validated fifteen GLI targets involved in MAPK and other signaling pathways. The manuscript is well prepared and presented. However, some concerns related to manuscript are given below.

1.     Out of the fourteen melanoma cell lines evaluated only four of them expressed elevated levels of GLI1, GLI2 and GLI3 proteins. It means less then 30% of the tested melanoma cell lines expressed elevated levels of these proteins. Some of the melanoma cell lines have weak or no expression of these proteins. The authors are encouraged to provide proper explanation in relation to their findings.

2.     Melanoma cell lines that are resistant to MAPK pathway inhibitors should have been used in these studies.

3.     The authors should have validated the in vitro findings by using in vivo models (Xenografts or PDXs).

Author Response

Reviewer 3

The authors in this manuscript performed studies to identify new Hedgehog-GLI (HH-GLI)-regulated targets in melanoma and its crosstalk with MAPK pathway. For these studies, the authors utilized RNA sequencing and ChIP sequencing, and further validated their findings by qPCR. Presented data in the manuscript indicate that the protein levels of GLI1, GLI2 and GLI3 not detected equally in tested melanoma cell lines with different genetic background. The authors further utilized the melanoma cell lines that expressed elevated levels of all three GLI proteins and further validated fifteen GLI targets involved in MAPK and other signaling pathways. The manuscript is well prepared and presented. However, some concerns related to manuscript are given below.

  1. Out of the fourteen melanoma cell lines evaluated only four of them expressed elevated levels of GLI1, GLI2 and GLI3 proteins. It means less then 30% of the tested melanoma cell lines expressed elevated levels of these proteins. Some of the melanoma cell lines have weak or no expression of these proteins. The authors are encouraged to provide proper explanation in relation to their findings.

Answer: Thank you for your comment. Although only four out of fourteen melanoma cell lines have expressed elevated levels of all three GLI proteins at the same time, it is visible from the Figure 1A that almost all of them (with an exception of HS895.SK) express receptor PTCH1, which is the main component of HH-GLI pathway. Also 12 out of 14 cell lines express significant levels of GLI3 and 10 cell lines express also some levels of GLI3 repressor form. As for the further experiments (RNA-seq and ChIP-seq) we have used only the three cell lines that have the best expression of all three GLI proteins (cell lines CHL-1, A375 and MEL224), so we consider our results reliable and thrustworthy.

  1. Melanoma cell lines that are resistant to MAPK pathway inhibitors should have been used in these studies.

Answer: The goal of this study was to identify the GLI targets in melanoma, and for this we selected three melanoma cell lines with different mutational backgrounds to exclude the potential mutation bias. We agree there are some publications that show that HH-GLI signaling pathway may be involved in the development of resistance, but this was not the main focus of this work. In this setting, only one of the cell lines (A375) could be made resistant to vemurafenib, which would not be sufficient to address the question of resistance, but this is an excellent direction that we hope to address in the future by studying the targets identified and validated in this study.

  1. The authors should have validated the in vitro findings by using in vivo models (Xenografts or PDXs).

Answer: We agree that in vivo validation is the final step for any in vitro study. However this was not financially nor experimentally feasible within the scope of our project. We strongly encourage our colleagues in the field who have more funds and experience to build on our findings and use the xenograft models.

Reviewer 4 Report

This manuscript aims to provide new potential targets to consider while designing melanoma targeted therapy.The manuscript is clear, relevant for the field and well structured overall.  Although this work is quite promising and copious, there are quite a few lacunae that should be addressed to help improve its overall contribution to understanding the HH-GLI signaling pathway.

1.The quality of all the images should be improved for easy reading.

2.Figure2B have obvious errors, there are have two bar to show the part shared by GL1, GL2 and GL3. This figure is recommended to use venn instead of upsetR, it will be more clear.

3.Authors need to elaborate on the effect of HH-GLI signaling pathway on melanoma in the introduction section instead of just passing it by in a single sentence.

4.Figure 1, although the authors add KRAS mutation as a control group, but the reason is not clearly in the introduction or in this section.

5.Authors should have clearly indicated GL1, GL2 and GL3 at the bottom of Figure 2D, the current chart is not clear

6.Line390, please check the sentence and add punctuation

Author Response

Reviewer 4

This manuscript aims to provide new potential targets to consider while designing melanoma targeted therapy.The manuscript is clear, relevant for the field and well structured overall.  Although this work is quite promising and copious, there are quite a few lacunae that should be addressed to help improve its overall contribution to understanding the HH-GLI signaling pathway.

1.The quality of all the images should be improved for easy reading.

Answer: Thank you for your suggestion. We have provided all images in high quality as PDF versions, because they are mostly vector graphics obtained by R (with an exception for western blot image and Gene expression plot in Figure 1). In the word document of the manuscript we can't embed the PDF versions, so the images in the document are lower quality. In the final version of the paper the high quality images should be avaliable for readers.

2.Figure2B have obvious errors, there are have two bar to show the part shared by GL1, GL2 and GL3. This figure is recommended to use venn instead of upsetR, it will be more clear.

Answer: Thank you for your comment. There are no errors in Figure 2B. This confusion is a result of the lower quality of the images embedded in the word document, and maybe because of the color (red and grey circles) used on the graph. If you open the PDF (High quality version of Figure 2) it is visible that the second bar does not have red circle in the middle, as the first bar. Only the first bar represents the common targets of all three GLI proteins.

3.Authors need to elaborate on the effect of HH-GLI signaling pathway on melanoma in the introduction section instead of just passing it by in a single sentence.

Answer: Thank you for your suggestion. We have inserted a few extra sentences explaining the effect of HH-GLI pathway in melanoma in the Introduction section (lines 48-51).

4.Figure 1, although the authors add KRAS mutation as a control group, but the reason is not clearly in the introduction or in this section.

Answer: Thank your for your comment. We did not add KRAS mutated cell line as a control. Figure 1 shows that we have tested the activation of HH-GLI signaling pathway on the whole panel of 14 melanoma cell lines by western blot, qPCR and MTT. Since we have only one KRAS mutated cell lines in our panel of melanoma cell lines we have used only this one and not more. We have also used this cell line during validation by qPCR to show if there are any differences in the results among all mutations (WT, BRASmut, NRASmut and KRASmut). We showed that mutation of the cell lines does not effect the target gene expression. In melanoma, KRAS mutation is very rare - only 2-3% (Al Mahi A, Ablain J, 2022) so we did not focus on this mutation in the Introduction.

5.Authors should have clearly indicated GL1, GL2 and GL3 at the bottom of Figure 2D, the current chart is not clear.

Answer: GLI1, GLI2 and GLI3 are indicated on top of the Figure 2D (depleted or enriched).

6.Line390, please check the sentence and add punctuation

Answer: We could not identify the error in the sentence you indicated.

Round 2

Reviewer 3 Report

None